# Methaemoglobin as a surrogate marker of primaquine antihypnozoite activity in *Plasmodium vivax* malaria: A systematic review and individual patient data meta-analysis

Ihsan Fadilah[1,2]*, Robert J. Commons[3,4,5], Nguyen Hoang Chau[6], Cindy S. Chu[2,7], Nicholas P. J. Day[2,8], Gavin C. K. W. Koh[9], Justin A. Green[10], Marcus VG Lacerda[11,12,13], Alejandro Llanos-Cuentas[14], Erni J. Nelwan[15,16], Francois Nosten[2,7], Ayodhia Pitaloka Pasaribu[17,18], Inge Sutanto[19], Walter R. J. Taylor[2,8], Kamala Thriemer[3], Ric N. Price[2,3,4,8], Nicholas J. White[2,8], J. Kevin Baird[1,2], James A. Watson[2,6,20]*

1 Oxford University Clinical Research Unit Indonesia, Faculty of Medicine Universitas Indonesia, Jakarta, Indonesia, 2 Centre for Tropical Medicine and Global Health, Nuffield Department of Medicine, University of Oxford, Oxford, United Kingdom, 3 Menzies School of Health Research, Charles Darwin University, Darwin, Australia, 4 WorldWide Antimalarial Resistance Network, Asia-Pacific Regional Hub–Australia, Melbourne, Australia, 5 General and Subspecialty Medicine, Grampians Health, Ballarat, Australia, 6 Oxford University Clinical Research Unit, Hospital for Tropical Diseases, Ho Chi Minh City, Vietnam, 7 Shoklo Malaria Research Unit, Mahidol Oxford Tropical Medicine Research Unit, Faculty of Tropical Medicine, Mahidol University, Mae Sot, Thailand, 8 Mahidol Oxford Tropical Medicine Research Unit, Faculty of Tropical Medicine, Mahidol University, Bangkok, Thailand, 9 Formerly Senior Director, Global Health, GlaxoSmithKline, Brentford, United Kingdom, 10 Department of Infectious Diseases, Northwick Park Hospital, Harrow, United Kingdom, 11 Fundação de Medicina Tropical Dr Heitor Vieira Dourado, Manaus, Brazil, 12 Instituto Leônidas e Maria Deane, Fiocruz, Manaus, Brazil, 13 University of Texas Medical Branch, Galveston, Texas, United States of America, 14 Universidad Peruana Cayetano Heredia, Instituto de Medicina Tropical Alexander von Humboldt, Unit of Leishmaniasis and Malaria, Lima, Peru, 15 Faculty of Medicine, Universitas Indonesia, Jakarta, Indonesia, 16 Division of Tropical Medicine and Infectious Disease, Department of Internal Medicine, Cipto Mangunkusumo Hospital, Jakarta, Indonesia, 17 Department of Pediatrics, Medical Faculty, Universitas Sumatera Utara, Medan, Indonesia, 18 Tridarma Healthcare Empowerment Foundation, Medan, Indonesia, 19 Department of Parasitology, Faculty of Medicine, Universitas Indonesia, Jakarta, Indonesia, 20 Infectious Diseases Data Observatory, Oxford, United Kingdom

* ifadilah@oucru.org (IF); jwatowatson@gmail.com (JAW)

## Abstract

### Background

The 8-aminoquinolines, primaquine and tafenoquine, are the only available drugs for the radical cure of *Plasmodium vivax* hypnozoites. Previous evidence suggests that there is dose-dependent 8-aminoquinoline induced methaemoglobinaemia and that higher methaemoglobin concentrations are associated with a lower risk of *P. vivax* recurrence. We undertook a systematic review and individual patient data meta-analysis to examine the utility of methaemoglobin as a population-level surrogate endpoint for 8-aminoquinoline antihypnozoite activity to prevent *P. vivax* recurrence.

### Methods and findings

We conducted a systematic search of Medline, Embase, Web of Science, and the Cochrane Library, from 1 January 2000 to 29 September 2022, inclusive, of prospective clinical

**Data Availability Statement:** Pseudonymised participant data used in this study can be accessed

via the WorldWide Antimalarial Resistance Network (wwarn.org). Requests for access will be reviewed by a data access committee to ensure that use of data protects the interests of the participants and researchers according to the terms of ethics approval and principles of equitable data sharing. Requests can be submitted by email to malariaDAC@iddo.org via the data access form available at https://www.wwarn.org/working-together/sharing-accessing-data/accessing-data. Code for data analysis and visualisation is available at https://github.com/ihsanfadil/methb7.

**Funding:** IF is supported by the Oxford Nuffield Department of Medicine Tropical Network Fund. RNP and RJC are supported by Australian National Health and Medical Research Council (NHMRC) Investigator Grants (2008501 and 1194701, respectively). Shoklo Malaria Research Unit (grant 220211) and FN (grant 089179) are supported by the Wellcome Trust. MVGL is a fellow from the National Council for Scientific and Technological Development (CNPq). KT is a CSL Centenary Fellow. JAW is a Sir Henry Dale Fellow funded by the Wellcome Trust (223253/Z/21/Z). NJW is a Wellcome Trust Principal Fellow (093956/Z/10/C). JKB and KT receive institutional research funding from Medicines for Malaria Venture. JKB reports GSK, Wellcome Trust, and Sanaria. This research was supported by grants from the Wellcome Trust. The sponsors or funders did not play any role in the study design, data collection and analysis, decision to publish, or preparation of the manuscript.

**Competing interests:** JAG and GCKWK are former employees of GSK and hold shares in GSK and AstraZeneca. GCKWK reports travel support from AstraZeneca. JKB reports participation on the US National Institutes of Health data safety monitoring board; and membership of the editorial board of Travel Medicine and Infectious Disease and the guidelines development group for malaria control and elimination, Global Malaria Programme, WHO. RJC, JKB, and RNP report contributions to Up-to-Date. All other authors declare no competing interests.

**Abbreviations:** ACT, artemisinin-combination therapies; aHR, adjusted hazard ratio; CI, confidence interval; DHA, dihydroartemisinin; G6PD, glucose-6-phosphate dehydrogenase; IQR, interquartile range; QUIPS, Quality in Prognosis Studies; WWARN, Worldwide Antimalarial Resistance Network.

efficacy studies of acute, uncomplicated *P. vivax* malaria mono-infections treated with radical curative doses of primaquine. The day 7 methaemoglobin concentration was the primary surrogate outcome of interest. The primary clinical outcome was the time to first *P. vivax* recurrence between day 7 and day 120 after enrolment. We used multivariable Cox proportional-hazards regression with site random-effects to characterise the time to first recurrence as a function of the day 7 methaemoglobin percentage (log base 2 transformed), adjusted for the partner schizonticidal drug, the primaquine regimen duration as a proxy for the total primaquine dose (mg base/kg), the daily primaquine dose (mg/kg), and other factors. The systematic review protocol was registered with PROSPERO (CRD42023345956).

We identified 219 *P. vivax* efficacy studies, of which 8 provided relevant individual-level data from patients treated with primaquine; all were randomised, parallel arm clinical trials assessed as having low or moderate risk of bias. In the primary analysis data set, there were 1,747 patients with normal glucose-6-phosphate dehydrogenase (G6PD) activity enrolled from 24 study sites across 8 different countries (Indonesia, Brazil, Vietnam, Thailand, Peru, Colombia, Ethiopia, and India). We observed an increasing dose-response relationship between the daily weight-adjusted primaquine dose and day 7 methaemoglobin level. For a given primaquine dose regimen, an observed doubling in day 7 methaemoglobin percentage was associated with an estimated 30% reduction in the risk of *P. vivax* recurrence (adjusted hazard ratio = 0.70; 95% confidence interval [CI] [0.57, 0.86]; $p = 0.0005$). These pooled estimates were largely consistent across the study sites. Using day 7 methaemoglobin as a surrogate endpoint for recurrence would reduce required sample sizes by approximately 40%. Study limitations include the inability to distinguish between recrudescence, reinfection, and relapse in *P. vivax* recurrences.

## Conclusions

For a given primaquine regimen, higher methaemoglobin on day 7 was associated with a reduced risk of *P. vivax* recurrence. Under our proposed causal model, this justifies the use of methaemoglobin as a population-level surrogate endpoint for primaquine antihypnozoite activity in patients with *P. vivax* malaria who have normal G6PD activity.

## Author summary

### Why was this study done?

- *Plasmodium vivax* causes recurrent malaria due to dormant liver stage parasites.

- The pro-drugs primaquine and tafenoquine are the only available treatments to prevent relapsing malaria, but their optimal dosing remains unclear.

- The active metabolites of primaquine and tafenoquine cause predictable increases in blood methaemoglobin; these same metabolites also kill liver parasites.

- This suggests that increases in blood methaemoglobin could be used as a population-level surrogate endpoint for liver stage parasite killing.

### What did the researchers do and find?

- We conducted an individual patient data meta-analysis examining the utility and validity of blood methaemoglobin as a surrogate endpoint for recurrent vivax malaria.

- We systematically reviewed and pooled the available methaemoglobin and clinical data from patients with *P. vivax* malaria treated with primaquine over the last 20 years.

- We fit statistical models to these data to quantify the relationship between day 7 methaemoglobin levels and the risk of recurrent *P. vivax* malaria.

- We observed an increasing dose-response relationship between weight-adjusted primaquine dose and methaemoglobin levels measured on day 7.

- We found that for a fixed primaquine dose regimen, a doubling in day 7 methaemoglobin was associated with an estimated 30% reduction in the risk of vivax recurrence.

- Using day 7 methaemoglobin levels as a surrogate endpoint for recurrent malaria could reduce sample sizes needed in future studies by about 40% and reduce follow-up duration by 94%.

### What do these findings mean?

- Patients with higher methaemoglobin levels a week after starting primaquine treatment appear to have a lower risk of recurrence.

- Day 7 methaemoglobin levels can potentially be used as a proxy for later vivax recurrence in exploratory trials, making it more efficient (e.g., fewer volunteers and resources required) to determine whether new drugs or regimens are effective.

- Study limitations include the inability to determine whether *P. vivax* recurrences are due to treatment failure, reinfection, or relapse.

## Introduction

The human malaria parasites *Plasmodium vivax* and *Plasmodium ovale* are characterised by their ability to form dormant liver stage parasites called hypnozoites, which activate weeks to months later to cause relapsing bloodstream infection [1]. *P. vivax* is the most geographically widespread cause of human malaria and is a major challenge in malaria elimination. Relapse contributes substantially to the overall burden of symptomatic vivax malaria, causing over 75% of all symptomatic infections [2]. Preventing relapse is crucial for eliminating the burden of vivax malaria morbidity and mortality.

The only drugs available for radical cure (killing latent hypnozoites) are primaquine and tafenoquine. Both are thought to be prodrugs [3,4], necessitating metabolic activation to produce hypnozonticidal activity. The precise mechanism and active metabolites of primaquine and tafenoquine are poorly characterised [5]. Patients with vivax malaria, who receive an equivalent 8-aminoquinoline dose per body weight, may variably metabolise the drug leading to varying risks of later relapse. Some of this variation in biotransformation is due to

polymorphisms in cytochrome *P450 (CYP) 2D6* [3]. Poorer metabolisers are associated with higher relapse rates [6,7].

Both primaquine and tafenoquine cause predictable increases in blood methaemoglobin which results from the action of their oxidative metabolites [8]. This involves a reversible increase in the conversion rate of intra-erythrocytic reduced ($Fe^{++}$) haem iron in haemoglobin to its oxidised ($Fe^{+++}$) form [8]. Following daily administration of primaquine or single dose administration of tafenoquine, blood methaemoglobin concentrations gradually increase, reaching peak concentrations after approximately 1 week [9]. It has been hypothesised that the same oxidative metabolites responsible for methaemoglobinaemia are also responsible for inducing haemolysis in glucose-6-phosphate dehydrogenase (G6PD) deficiency, killing mature gametocytes of *P. falciparum*, and killing liver stage hypnozoites [8]. Early experiments with primaquine and its analogues indicated that there were greater increases in blood methaemoglobin for molecules which had improved radical-cure efficacy [8]. Two recent pharmacometric studies estimated an adjusted proportional reduction in the risk of vivax recurrence of approximately 10% for primaquine [10] and 20% for tafenoquine [4] for each additional percentage-point increase in day 7 methaemoglobin. Based on these results, it is hypothesised that increases in methaemoglobin may serve as a proxy for 8-aminoquinoline antihypnozoite activity and, as such, a potential surrogate endpoint for clinical trials to quantify the antirelapse efficacy of 8-aminoquinoline drugs in vivax malaria.

A surrogate endpoint is a patient characteristic, such as a biomarker, intended to substitute for a clinical outcome [11]—specifically vivax recurrence in this context. To validate surrogacy, high-quality studies need to demonstrate that a putative biomarker is affected by the drug intervention and that drug-induced change in the biomarker level can predict the effect on the outcome of interest [12,13]. We examined the utility of methaemoglobin as a population-level surrogate endpoint for vivax recurrence in a pooled individual patient data meta-analysis using available data from *P. vivax* patients treated with primaquine.

## Methods

### Search strategy and selection criteria

We conducted a systematic search of Medline, Embase, Web of Science, and the Cochrane Library based on an existing living systematic review [14] of prospective clinical efficacy studies of acute, uncomplicated *P. vivax* malaria mono-infections treated with primaquine published between 1 January 2000 and 29 September 2022 in any language. Studies were eligible for the individual patient data meta-analysis if they were randomised therapeutic trials and prospective cohort studies with a minimum active follow-up of 42 days that recorded methaemoglobin data (at baseline and at least once in the first week of follow-up between day 5 and day 9) following daily primaquine administration given over multiple days. The systematic review was subsequently updated to 26 July 2024 but 2 potentially eligible clinical trials published between 30 September 2022 and 26 July 2024 were not included in the individual patient data meta-analysis due to time constraints in obtaining approval and standardising these data. Only individuals with normal G6PD levels (i.e., ≥30% G6PD activity or a negative qualitative test) were included in the analysis. In individuals with G6PD deficiency, primaquine administration does not lead to the same methaemoglobin increases [15].

Studies were included if the primaquine regimen was administered within the first 3 days of schizonticidal treatment. Search terms are provided in Supporting information (Systematic search terms for the databases in S1 Text). This systematic review was conducted by 2 reviewers (IF and RJC), with discrepancies resolved through discussion. The review protocol was registered with PROSPERO (CRD42023345956).

## Data pooling

The corresponding authors and/or principal investigators of eligible studies that met the study criteria were invited through direct email to contribute their individual patient data. Relevant data from unpublished studies were requested wherever possible. Shared data were uploaded to the Worldwide Antimalarial Resistance Network (WWARN) repository for curation and standardisation, utilising the IDDO SDTM Implementation Guide [16]. We excluded individual patients with missing information on age, sex, body weight, baseline parasite density, primaquine regimen, or schizonticidal treatment. Patients with severe malaria, pregnancy, mixed-species infection, or those who received adjunctive antimalarials after the initial schizonticidal treatment were also excluded.

All studies included in our meta-analysis provided pseudonymised individual data and had obtained ethical approvals from the corresponding site of origin. Therefore, additional ethical approval was not required for the current analysis, as per the Oxford Tropical Research Ethics Committee. We adhered to the PRISMA-IPD guidelines [17] for reporting this systematic review and individual patient data meta-analysis (Table A in S1 Text).

## Outcomes

The primary clinical outcome was the time to first *P. vivax* recurrence (i.e., any episode of *P. vivax* parasitaemia irrespective of symptoms) between day 7 (the starting point for prediction and follow-up) and day 120 [4] after the initial (i.e., first dose) primaquine administration for the initial vivax malaria illness.

The secondary outcomes were the binary outcome of any *P. vivax* recurrence between day 7 and day 120 after starting primaquine and the maximum absolute change in haemoglobin concentration from day 0 (day of randomisation) to days 2 to 3 (expected days of the lowest haemoglobin level [18]) following the start of any antimalarial treatment. In addition to being the main predictor of interest in the statistical model, we also specified the day 7 methaemoglobin concentration as an outcome and modelled this biomarker as a function of daily mg (base) per kilogram primaquine dose.

## Data analysis

We presented study-level summary statistics to highlight sample characteristics and potential heterogeneity across the included studies. The daily distributions of methaemoglobin levels, stratified by schizonticidal drug and primaquine regimen (low total dose 14 day, high total dose 7 day), were plotted to illustrate the temporal dynamics of primaquine-induced methaemoglobin production during radical cure treatment.

The primary predictor of interest (surrogate outcome) was the day 7 methaemoglobin concentration, expressed as a percentage of the total haemoglobin concentration. Day 7 was pre-specified and typically, methaemoglobin concentrations peak after approximately a week of commencing the daily primaquine regimens. All studies measured methaemoglobin by transcutaneous pulse CO-oximetry. The day 7 methaemoglobin percentage (log base 2 transformed) was included in the statistical model as a continuous variable. If the day 7 methaemoglobin percentage was recorded as 0, we replaced it with 1, assuming an approximately physiological methaemoglobin level of 1%. Zero recordings are likely to represent misreadings of the analytical machine.

We proposed a causal directed acyclic graph for this analysis to guide model specification and aid interpretation of results (Fig 1). Missing day 7 methaemoglobin percentages were linearly imputed using levels measured within ±2 days. If only 1 measurement was available, then the imputation assumed a constant (i.e., the single value observed was used). If no

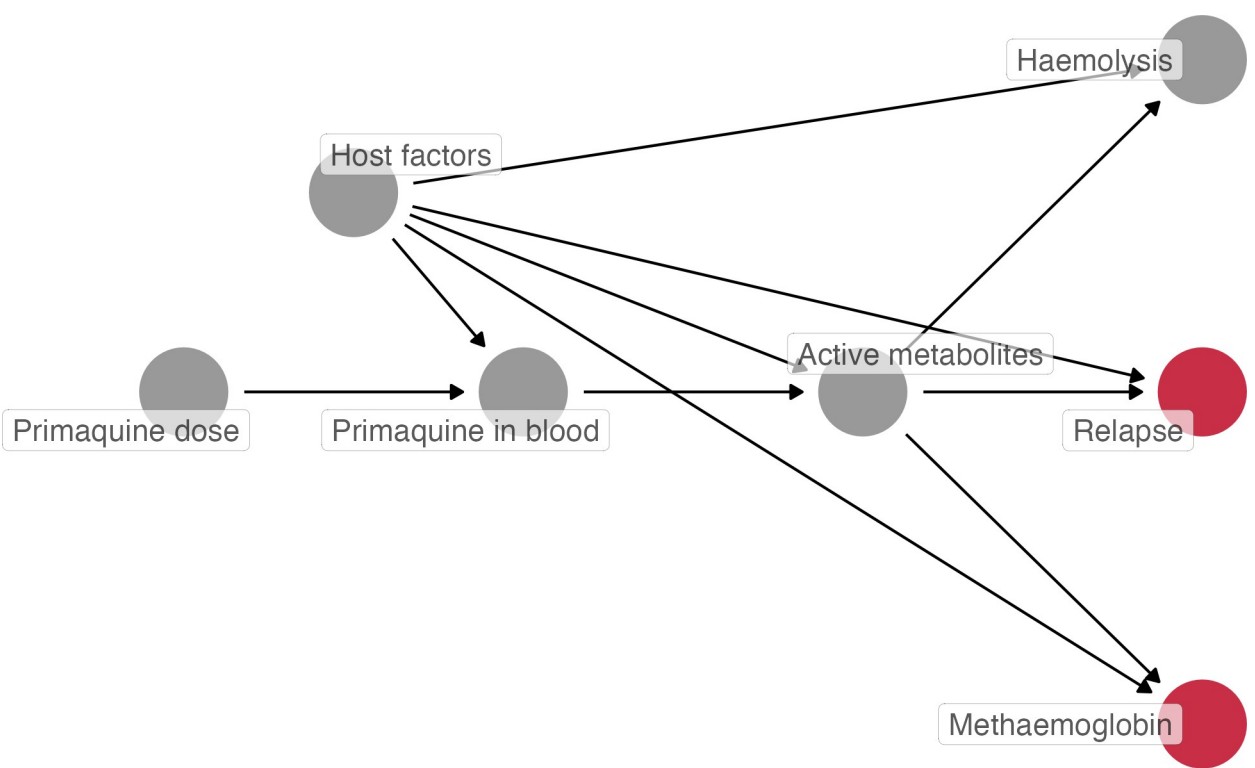

**Fig 1. Directed acyclic graph showing our hypothesised causal relationships between primaquine-induced changes in blood methaemoglobin concentrations and *P. vivax* relapse.** Red nodes represent the outcomes of interest: relapse and blood methaemoglobin (measured on day 7, for example), between which the association was estimated. Under this causal model, blood methaemoglobin is a proxy measurement for the hypnozontocidal activity of primaquine (but not on the causal pathway mediating the effect of primaquine on relapse). Host factors include but not limited to patient's genetics (e.g., those related to *CYP2D6* and *G6PD*), behaviours, age, immunity to *P. vivax*, and geographical location.

measurements were available within this timeframe (day 5 to day 9), the patient was excluded from the analysis.

In the main analysis, patients were right censored at the time of the first recurrent vivax parasitaemia (outcome), any malaria parasitaemia, loss to follow up, blood smear gap of >60 days, or the last day of study, whichever occurred first. We used multivariable, random-effects Cox proportional-hazards regression to model the time to first recurrence as a function of the day 7 methaemoglobin percentage ($\log_2$ transformed) under a one-stage individual patient data meta-analysis framework. This model adjusted for daily mg/kg primaquine dose, primaquine duration (a proxy for total mg/kg primaquine dose), within-site and across-site linear interactions [19] between daily mg/kg primaquine dose and primaquine duration, age, sex, schizonticidal drug, and baseline parasite density (natural-log transformed). A random intercept and a random slope for day 7 methaemoglobin concentration were included to account for between-site effect-heterogeneity. Linearity and proportional-hazards assumptions were checked. The adjusted hazard ratio can be interpreted as the estimated predictive effect of each doubling in day 7 methaemoglobin percentage, over and above the adjustment factors.

To compare our estimates with those from previous studies [4,10], we also specified the day 7 methaemoglobin concentration on its original scale and refitted the main survival analysis model. Additionally, we separately (by study and primaquine regimen) fit a more parsimonious Cox proportional-hazards model that adjusted for daily mg/kg primaquine dose and pooled the estimates obtained from all the study-regimen categories using a two-stage individual-patient data meta-analysis approach [20]. A similar model specification to the one-stage

approach that included a few more adjustment factors was not possible as the data were sparse (i.e., few recurrences). A forest plot was constructed to visualise the results under the common-effect and random-effects models. Assuming that the primaquine regimen was the primary determinant of variation in the outcome, dependence in the estimates derived from the same study [9,21] was expected to be minimal.

We estimated the adjusted predictive effect of the day 7 methaemoglobin percentage ($\log_2$ transformed) on the odds of vivax recurrence using multivariable, random-effects binary logistic regression. This model was limited to patients with at least 120 days of follow-up and adjusted for daily mg/kg primaquine dose, primaquine duration, within-site and across-site linear interactions between daily mg/kg primaquine dose and primaquine duration. A random intercept for study site was specified. The association between the maximum absolute change in haemoglobin concentration from day 0 to days 2 to 3 and the day 7 methaemoglobin percentage was estimated using multivariable, random-effects linear regression. This model included baseline haemoglobin concentration, daily mg/kg primaquine dose, age, sex, schizonticidal drug, baseline parasite density (natural-log transformed) as common-effect covariates, and a random intercept and slope for study site and daily mg/kg primaquine dose, respectively. This model was restricted to patients who started primaquine treatment on day 0. If a haemoglobin measurement was missing, haematocrit was used to impute the haemoglobin concentration using the formula derived from a large series of malaria patients; haemoglobin = (haematocrit − 5.62) ÷ 2.60, where haematocrit was measured in percent and haemoglobin was measured in grams per decilitre [22]. If haematocrit remained missing, these patients were excluded. We also estimated the association of daily mg/kg primaquine dose and day 7 methaemoglobin percentage using a random-effects linear model, allowing for a random intercept and slope for study site and daily mg/kg primaquine dose, respectively.

We provide illustrative sample-size calculations to demonstrate how our findings could contribute to making future studies of drug discovery or regimen optimisation in *P. vivax* more efficient by using blood methaemoglobin as a surrogate outcome. We estimated that a 0.5-mg/kg increase in daily primaquine dose results in a 0.39 increase in the $\log_2$ day 7 methaemoglobin (i.e., a 30% increase). We estimated the standard deviation of the $\log_2$ day 7 methaemoglobin level, conditional on the daily mg/kg primaquine dose from the pooled data. For example, the standard deviation for the high daily dose group is approximately 1.19. Assuming a normal distribution for the $\log_2$ day 7 methaemoglobin conditional on the daily dose allows for a simple calculation of the required sample size. This is derived from a *t* test for a difference between 2 normal distributions with mean difference (effect size) of 0.39 and equal standard deviations in both groups. For the power calculation for the clinical endpoints, we assumed recurrence risks of 16% versus 8% (corresponding to primaquine doses of 0.5 versus 1 mg/kg over 7 days) [23].

Risk of bias related to individual studies was evaluated using the Quality in Prognosis Studies (QUIPS) tool [24] adapted to the current analysis (signalling questions for risk of bias assessment using the QUIPS tool adapted to the current analysis in S1 Text). Statistical analysis followed a prespecified plan [25] and was conducted using R Statistical Software (version 4.3.0). Statistical modelling was implemented using the R packages survival [26], coxme [27], metafor [28], rms [29], and lme4 [30].

## Results

### Study and patient selection

We identified 219 *P. vivax* efficacy studies published between 1 January 2000 and 29 September 2022 (Fig 2). After review, 206 studies were excluded (almost all because they did not measure methaemoglobin or did not have a primaquine arm which started in the first 3 days of

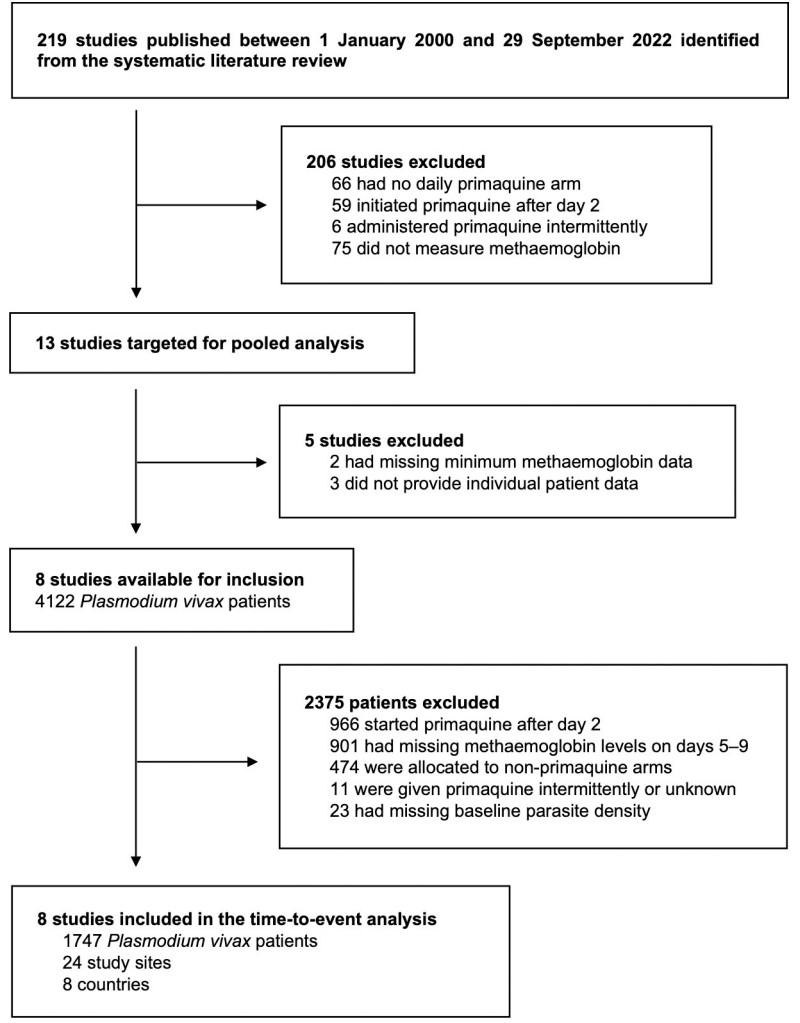

**Fig 2. Study and patient selection.** Databases systematically searched were from Medline, Embase, Web of Science, and the Cochrane Library. Patients included in the secondary analyses were subsets of the patients in the primary, time-to-event analysis.

treatment), leaving 13 studies eligible for the pooled analysis. Eight of these studies (8/13, 62%) provided individual-level data from patients treated with primaquine; all were randomised, parallel arm clinical trials and assessed as having low risk of bias (Table B in S1 Text). Individual-level data were available for 4,122 patients from these 8 trials, of whom 1,747 (42%) satisfied the inclusion and exclusion criteria for our analysis (primary data set). These patients were enrolled from 24 study sites across 8 different countries (Indonesia, Brazil, Vietnam, Thailand, Peru, Colombia, Ethiopia, and India; Fig A in S1 Text). Most patients had a follow-up of at least 120 days since commencing primaquine radical cure treatment (1,344, 77%–secondary sub-data set) and had haemoglobin concentration (or haematocrit) measured on day 0 of antimalarial treatment (1,360, 78%–haemolysis sub-data set).

## Patient characteristics

In the primary analysis (*n* = 1,747 patients), the median age was 20 years (interquartile range [IQR]: 12 to 32), with 89 patients (5.1%) younger than 5 years. Overall, most patients were

male (1,116, 64%), resided in the Asia-Pacific region (1,620, 93%), and were from locations with frequent *P. vivax* relapse periodicity (i.e., a median interval from the first acute episode to relapse of less than 47 days [31]; 1,614, 92%) and moderate transmission intensity (i.e., 1 to 9 cases per 1,000 person-years [32]; 1,325, 76%). The majority of patients (1,138, 65%) were treated with artemisinin-combination therapies (ACT), started primaquine on day of enrolment (day 0; 1,496, 86%), and took daily primaquine over 14 days (1,194, 68%). The overall median daily-dose of primaquine was 0.52 mg/kg (IQR: 0.38 to 0.95, Fig B in S1 Text shows the weight-adjusted dose distribution by primaquine duration). Primaquine administration was fully supervised in most studies (i.e., all doses were directly observed; 1,571, 90%). Further details on baseline patient characteristics are summarised in Table 1. Eligible studies that were not included [33–37] in the analysis tended to have a shorter duration of follow-up but were otherwise similar in other characteristics (Tables C–E in S1 Text).

### Effect of primaquine on methaemoglobin concentrations

There were consistent increases in methaemoglobin concentration from baseline to day 7 during primaquine treatment for both the 7-day and 14-day regimens. On average, the peak observed methaemoglobin concentrations occurred on day 7 following the start of primaquine administration, with a median day 7 methaemoglobin level of 6.0% (IQR: 3.3 to 9.0). Subsequently, methaemoglobin levels tended to decrease slowly in the 7-day or plateau in the 14-day regimen groups (Figs 3 and C in S1 Text).

Among patients treated with a low-to-intermediate daily primaquine dose, day 7 methaemoglobin was lower in the studies where primaquine was combined with chloroquine or quinine as a partner drug compared with ACT (Fig D in S1 Text). However, there was no clear evidence suggesting a drug–drug interaction at the patient level as there was insufficient within-site variation in the pooled data (within-site interaction, $p = 0.29$; across-site interaction, $p = 0.028$). The highest levels were observed among adolescent patients. Younger and older patients had lower primaquine-induced methaemoglobinaemia (Fig E in S1 Text). Fig F in S1 Text shows the distribution of day 7 methaemoglobin concentrations by primaquine regimen.

There was dose-dependent primaquine-induced methaemoglobin production. On average, for every additional 0.1 mg/kg increase in the daily primaquine dose, there was an associated 0.34 percentage-point increase in the day-7 methaemoglobin concentration (95% confidence interval [CI] = [0.16, 0.52]; $p = 0.002$). This effect remained consistent across all sites (between-site standard deviation in mean difference = 0.20; this quantifies the variability in the estimated effect of the daily primaquine dose on the day-7 methaemoglobin concentration across different sites, Fig G in S1 Text shows the study site-specific random-effects estimates). For a particular mg/kg daily dose, patients who later developed vivax recurrences (largely attributable to relapses) during follow-up had lower day 7 methaemoglobin values (Fig 4). In addition to the heterogeneous background risk of relapse across sites, we observed significant interindividual variation in the methaemoglobin response to primaquine treatment.

### Association of day 7 methaemoglobin concentrations with the risk of vivax recurrence

After adjusting for the daily and total dose of primaquine and other covariates, a doubling in the observed or imputed day 7 methaemoglobin percentage was associated with an estimated 30% reduction in the risk of vivax recurrence (adjusted hazard ratio [aHR] = 0.70; 95% CI [0.57, 0.86]; $p = 0.0005$. Table F in S1 Text shows an excerpt of the regression output). These pooled estimates were largely consistent across the study sites (between-site standard deviation

**Table 1. Demographic and patient characteristics at baseline.**

| Characteristic | Overall | Study | | | | | | | |
|---|---|---|---|---|---|---|---|---|---|
| | | Pasaribu 2013 [51] | Sutanto 2013 [47] | Llanos-Cuentas 2014 [44] | Nelwan 2015 [48] | Chu 2019 [9] | Lacerda 2019 [43] | Llanos-Cuentas 2019 [45] | Taylor 2019 [21] |
| Number of patients | 1,747 | 303 | 38 | 50 | 120 | 578 | 42 | 84 | 532 |
| **Region** | | | | | | | | | |
| Asia-Pacific | 1,620 (93%) | 303 (100%) | 38 (100%) | 22 (44%) | 120 (100%) | 578 (100%) | 4 (9.5%) | 23 (27%) | 532 (100%) |
| Americas | 124 (7.1%) | 0 (0%) | 0 (0%) | 28 (56%) | 0 (0%) | 0 (0%) | 35 (83%) | 61 (73%) | 0 (0%) |
| Africa | 3 (0.2%) | 0 (0%) | 0 (0%) | 0 (0%) | 0 (0%) | 0 (0%) | 3 (7.1%) | 0 (0%) | 0 (0%) |
| **Relapse periodicity$^{§}$** | | | | | | | | | |
| Low | 133 (7.6%) | 0 (0%) | 0 (0%) | 34 (68%) | 0 (0%) | 0 (0%) | 38 (90%) | 61 (73%) | 0 (0%) |
| High | 1,614 (92%) | 303 (100%) | 38 (100%) | 16 (32%) | 120 (100%) | 578 (100%) | 4 (9.5%) | 23 (27%) | 532 (100%) |
| **Transmission intensity$^{#}$** | | | | | | | | | |
| Low | 262 (15%) | 0 (0%) | 0 (0%) | 16 (32%) | 120 (100%) | 0 (0%) | 4 (9.5%) | 17 (20%) | 105 (20%) |
| Moderate | 1,325 (76%) | 303 (100%) | 0 (0%) | 6 (12%) | 0 (0%) | 578 (100%) | 0 (0%) | 11 (13%) | 427 (80%) |
| High | 160 (9.2%) | 0 (0%) | 38 (100%) | 28 (56%) | 0 (0%) | 0 (0%) | 38 (90%) | 56 (67%) | 0 (0%) |
| **Median age (years)** | 20 (12, 32) | 13 (9, 25) | 27 (25, 29) | 34 (26, 46) | 28 (25, 31) | 20 (13, 32) | 38 (24, 47) | 36 (25, 50) | 15 (10, 29) |
| **Age (years)** | | | | | | | | | |
| <5 | 89 (5.1%) | 26 (8.6%) | 0 (0%) | 0 (0%) | 0 (0%) | 24 (4.2%) | 0 (0%) | 0 (0%) | 39 (7.3%) |
| ≥5 and <15 | 534 (31%) | 144 (48%) | 0 (0%) | 0 (0%) | 0 (0%) | 168 (29%) | 0 (0%) | 0 (0%) | 222 (42%) |
| ≥15 | 1,124 (64%) | 133 (44%) | 38 (100%) | 50 (100%) | 120 (100%) | 386 (67%) | 42 (100%) | 84 (100%) | 271 (51%) |
| **Sex, male** | 1,116 (64%) | 169 (56%) | 38 (100%) | 35 (70%) | 120 (100%) | 363 (63%) | 30 (71%) | 52 (62%) | 309 (58%) |
| **Body weight (kg)** | 48 (30, 59) | 35 (21, 50) | 65 (59, 72) | 59 (49, 68) | 69 (63, 74) | 47 (33, 54) | 64 (55, 72) | 63 (55, 70) | 43 (24, 54) |
| **Schizonticidal drug** | | | | | | | | | |
| Artesunate/Amodiaquine | 146 (8.4%) | 146 (48%) | 0 (0%) | 0 (0%) | 0 (0%) | 0 (0%) | 0 (0%) | 0 (0%) | 0 (0%) |
| Artesunate/Pyronaridine | 60 (3.4%) | 0 (0%) | 0 (0%) | 0 (0%) | 60 (50%) | 0 (0%) | 0 (0%) | 0 (0%) | 0 (0%) |
| Dihydroartemisinin/ Piperaquine | 932 (53%) | 157 (52%) | 0 (0%) | 0 (0%) | 60 (50%) | 290 (50%) | 0 (0%) | 0 (0%) | 425 (80%) |
| Quinine | 38 (2.2%) | 0 (0%) | 38 (100%) | 0 (0%) | 0 (0%) | 0 (0%) | 0 (0%) | 0 (0%) | 0 (0%) |
| Chloroquine | 571 (33%) | 0 (0%) | 0 (0%) | 50 (100%) | 0 (0%) | 288 (50%) | 42 (100%) | 84 (100%) | 107 (20%) |
| **Primaquine duration** | | | | | | | | | |
| 7 days | 553 (32%) | 0 (0%) | 0 (0%) | 0 (0%) | 0 (0%) | 289 (50%) | 0 (0%) | 0 (0%) | 264 (50%) |
| 14 days | 1,194 (68%) | 303 (100%) | 38 (100%) | 50 (100%) | 120 (100%) | 289 (50%) | 42 (100%) | 84 (100%) | 268 (50%) |
| **Primaquine start** | | | | | | | | | |
| Day 0 | 1,496 (86%) | 303 (100%) | 27 (71%) | 0 (0%) | 120 (100%) | 511 (88%) | 3 (7.1%) | 0 (0%) | 532 (100%) |
| Day 1 | 181 (10%) | 0 (0%) | 9 (24%) | 49 (98%) | 0 (0%) | 0 (0%) | 39 (93%) | 84 (100%) | 0 (0%) |
| Day 2 | 70 (4.0%) | 0 (0%) | 2 (5.3%) | 1 (2.0%) | 0 (0%) | 67 (12%) | 0 (0%) | 0 (0%) | 0 (0%) |
| **Median primaquine daily dose (mg/kg)** | 0.52 (0.38, 0.95) | 0.31 (0.27, 0.34) | 0.51 (0.47, 0.59) | 0.25 (0.22, 0.30) | 0.53 (0.47, 0.60) | 0.87 (0.50, 1.00) | 0.23 (0.21, 0.27) | 0.25 (0.23, 0.29) | 0.66 (0.53, 1.03) |

(*Continued*)

**Table 1.** (Continued)

| Characteristic | Overall | Study | | | | | | | |
|---|---|---|---|---|---|---|---|---|---|
| | | Pasaribu 2013 [51] | Sutanto 2013 [47] | Llanos-Cuentas 2014 [44] | Nelwan 2015 [48] | Chu 2019 [9] | Lacerda 2019 [43] | Llanos-Cuentas 2019 [45] | Taylor 2019 [21] |
| **Primaquine daily dose (mg/kg)** | | | | | | | | | |
| <0.375 | 436 (25%) | 253 (83%) | 0 (0%) | 49 (98%) | 0 (0%) | 4 (0.7%) | 42 (100%) | 83 (99%) | 5 (0.9%) |
| ≥0.375 and <0.75 | 760 (44%) | 50 (17%) | 38 (100%) | 1 (2.0%) | 120 (100%) | 282 (49%) | 0 (0%) | 1 (1.2%) | 268 (50%) |
| ≥0.75 | 551 (32%) | 0 (0%) | 0 (0%) | 0 (0%) | 0 (0%) | 292 (51%) | 0 (0%) | 0 (0%) | 259 (49%) |
| **Primaquine dose calculation** | | | | | | | | | |
| Actual dosing | 1,324 (76%) | 0 (0%) | 38 (100%) | 50 (100%) | 0 (0%) | 578 (100%) | 42 (100%) | 84 (100%) | 532 (100%) |
| Protocol dosing | 423 (24%) | 303 (100%) | 0 (0%) | 0 (0%) | 120 (100%) | 0 (0%) | 0 (0%) | 0 (0%) | 0 (0%) |
| **Primaquine supervision** | | | | | | | | | |
| Fully supervised | 1,571 (90%) | 303 (100%) | 38 (100%) | 0 (0%) | 120 (100%) | 578 (100%) | 0 (0%) | 0 (0%) | 532 (100%) |
| Partially supervised | 176 (10%) | 0 (0%) | 0 (0%) | 50 (100%) | 0 (0%) | 0 (0%) | 42 (100%) | 84 (100%) | 0 (0%) |
| **Parasite density (asexual parasites/μl)** | 2,515 (617, 7,515) | 760 (320, 2,980) | 2,928 (784, 4,496) | 4,631 (1,993, 8,580) | 872 (144, 2,464) | 4,517 (1,440, 13,565) | 3,471 (1,589, 8,348) | 5,079 (1,913, 12,700) | 2,215 (551, 7,500) |
| **Haemoglobin (g/dl)** | 12.40 (11.30, 13.70) | 11.90 (10.80, 12.80) | 14.20 (13.10, 14.67) | NA (NA, NA) | NA (NA, NA) | 12.50 (11.40, 13.70) | NA (NA, NA) | NA (NA, NA) | 12.65 (11.50, 13.90) |
| Number of missing data | 212 | 0 | 38 | 28 | 120 | 0 | 11 | 15 | 0 |
| **Day-7 methaemoglobin (%)** | 6.0 (3.3, 9.0) | 3.8 (2.6, 5.6) | 5.6 (4.1, 7.5) | 2.5 (1.5, 6.0) | 5.5 (3.5, 8.2) | 6.5 (4.0, 8.9) | 2.4 (1.5, 6.1) | 1.4 (1.2, 7.3) | 7.1 (3.6, 11.0) |
| Number of missing data | 245 | 9 | 3 | 38 | 6 | 68 | 37 | 77 | 7 |
| **Manufacturer of primaquine tablets** | – | Phapros | Shin Poon | Sanofi | Sanofi | Thai Government | Sanofi | Sanofi | Centurion Laboratories |

Numbers are in median (first quartile, third quartile) or frequency (percentage). NA not available, g gram, dl decilitre, mg milligram, μl microlitre, kg kilogram.

§ Relapse periodicity was categorised as high (median relapse periodicity of 47 days or less) and low (median relapse periodicity of more than 47) [31].

# Transmission intensity was categorised as low (<1 case per 1,000 person-years), moderate (1 case to <10 cases per 1,000 person-years), and high (≥10 cases per 1,000 person-years) according to subnational malaria incidence estimates for the median year of study enrolment [32]. Missing day 7 methaemoglobin were linearly imputed with methaemoglobin data on ±2 days. Patients with missing haemoglobin or haematocrit data were excluded for the haemolysis sub-data set.

in aHR = 1.01; this is a multiplicative factor reflecting the variability in the estimated predictive-effect of the day-7 methaemoglobin concentration on the hazard across different study sites, Fig H in S1 Text shows the study-site specific random-effects estimates). There was no evidence of proportional hazards violations within the 120 days of follow-up. A sensitivity analysis restricted to patients with observed day 7 methaemoglobin values gave similar estimates (aHR = 0.66; 95% CI [0.52, 0.84]; $p$ = 0.0008; $n$ = 1,502). Controlling primaquine daily dose and duration, the relationship between day 7 methaemoglobin and the risk of vivax recurrence was generally consistent between the studies (Fig 5). A sensitivity analysis using a two-stage approach estimated a slightly smaller effect (dose-adjusted HR for vivax recurrence of 0.81 for each doubling in day 7 methaemoglobin percentage; 95% CI [0.67, 0.98]; $p$ = 0.037). This numerically different value estimated by the two-stage approach was primarily a result of less flexibility in model specification (e.g., fewer adjustment factors allowed to avoid estimation issues) and sparsity of data.

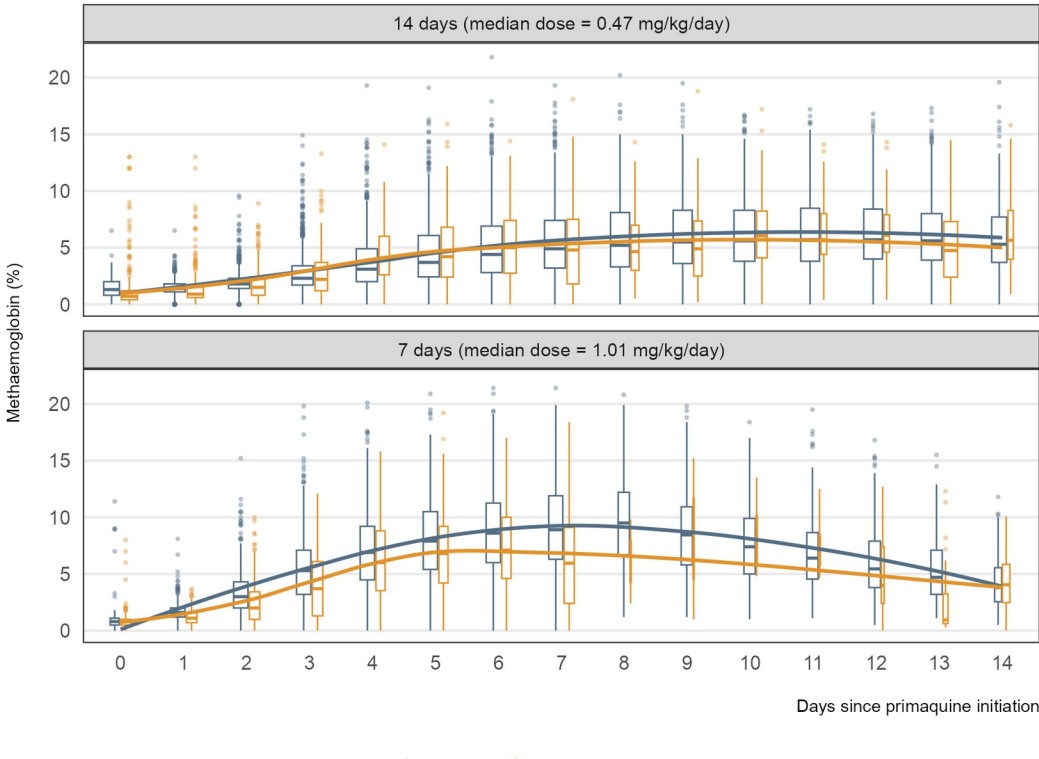

**Fig 3. Dynamics of primaquine-induced increases in blood methaemoglobin over time, stratified by primaquine regimen and schizonticidal drug.** Methaemoglobin levels increased after starting primaquine in both regimens, usually reaching a maximum after about a week. Methaemoglobin increased at a faster rate among the 7-day regimen reflecting the higher daily dose taken and generally methaemoglobin started to decrease during the second week, when primaquine was no longer administered. Meanwhile, after peaking at also day 7 for the 14-day regimen, methaemoglobin appears to be at a more constant level during the second week. Boxplot represents the distribution of methaemoglobin levels on a particular day following primaquine. In a boxplot, the box shows the interquartile range with a line for the median, and the whiskers extend to data points within 1.5 times the interquartile range, highlighting potential outliers. Solid curve summarises the data points by partner drug over time. Box width is proportional to the square root of the number of patients. ACT artemisinin-based combination therapy (artesunate/amodiaquine, artesunate/pyronaridine, dihydroartemisinin/piperaquine).

On the absolute scale (i.e., percentage of the total haemoglobin), each additional percentage-point increase in day 7 methaemoglobin was associated with an estimated 10% reduction in the risk of vivax recurrence (aHR = 0.90; 95% CI [0.84, 0.96]; $p = 0.003$). Improved model-fit ($p < 0.0001$) was observed by specifying day 7 methaemoglobinaemia on the logarithmic scale (as in the main model), indicating linearity between multiplicative changes in day-7 methaemoglobin level and the log-hazard ratio for vivax recurrence.

## Additional analyses

Of the 1,344 (77%) patients followed for at least 120 days, we observed a comparable relationship between day 7 methaemoglobin and the risk of any observed vivax recurrence (dose-adjusted odds ratio of 0.66 for each doubling in day 7 methaemoglobin; 95% CI [0.52, 0.83]; $p = 0.0004$). Of the 1,360 (78%) patients with normal G6PD activity with haemoglobin (or haematocrit) measured at baseline, there was little or no evidence of an association between the maximum absolute decrease in haemoglobin concentration from day 0 to days 2 to 3 and the day-7 methaemoglobinaemia (adjusted mean difference = 0.01; 95% CI [−0.21, 0.23], $p = 0.90$).

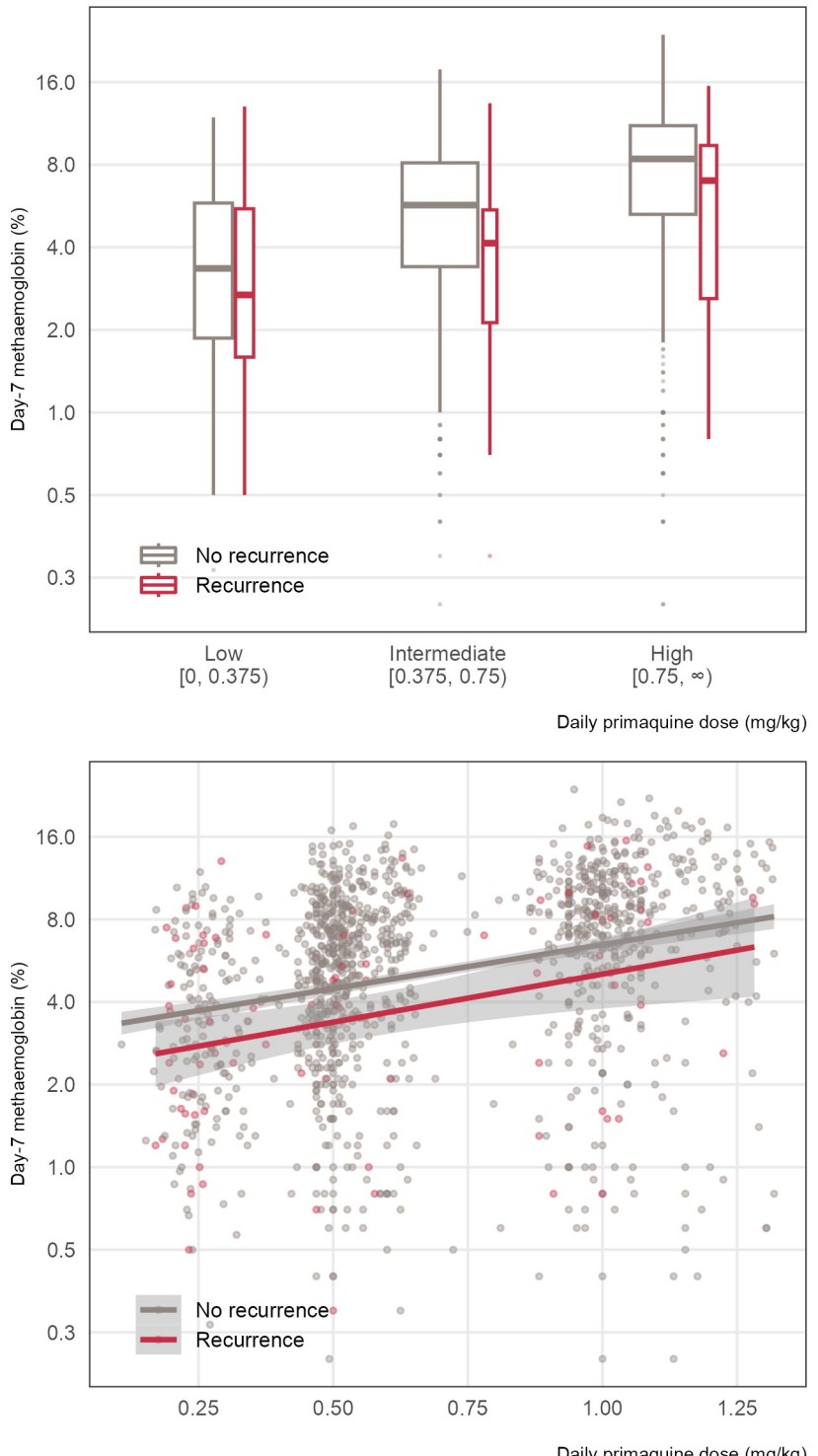

**Fig 4. Day 7 blood methaemoglobin (%) as a function of daily mg/kg primaquine dose and *P. vivax* recurrence status.** There was an increasing trend of day 7 methaemoglobin as daily primaquine dose increased among patients with at least 120 days of follow up. Patients who developed *P. vivax* recurrences typically had lower day 7 methaemoglobin levels. Top panel highlights comparisons between the dose groups and recurrence status using percentile summaries for day 7 methaemoglobin levels. Bottom panel provides a more detailed view of individual variability in day 7 methaemoglobin levels across different daily primaquine doses received and recurrence statuses. Solid line on the bottom panel denotes a regression line (with a 95% CI) of day-7 methaemoglobin levels on daily

primaquine dose. Vertical axis is shown on the logarithmic scale. Box width is proportional to the square root of the number of patients. In a boxplot, the box shows the interquartile range with a line for the median, and the whiskers extend to data points within 1.5 times the interquartile range, highlighting potential outliers (dots).

Assuming a comparison between 2 primaquine doses whereby the higher dose results in half the risk of vivax recurrence with recurrence rates of 16% versus 8% (corresponding to primaquine doses of 0.5 versus 1 mg/kg over 7 days), a two-arm randomised trial aiming to show superiority would require a sample size of 256 individuals per group to achieve 80% power with a 5% false positive rate. In contrast, using day 7 methaemoglobin as a surrogate endpoint for vivax recurrence, the required sample size would be reduced by approximately 42% ($n = 148$ individuals per group) for high daily dose primaquine and could be even more for lower daily doses (Fig I in S1 Text).

## Discussion

In this systematic review and individual patient data meta-analysis, we confirmed that higher primaquine-induced methaemoglobin concentrations on day 7 are associated with lower rates of *P. vivax* recurrence [8]. This finding remained consistent across diverse study sites with varying populations and levels of transmission intensity. Our analysis found no indication of a differential predictive effect of day 7 methaemoglobin between the 7-day and 14-day primaquine regimens. Additionally, we observed a positive dose-response relationship between the daily weight-adjusted primaquine dose and day 7 methaemoglobin level.

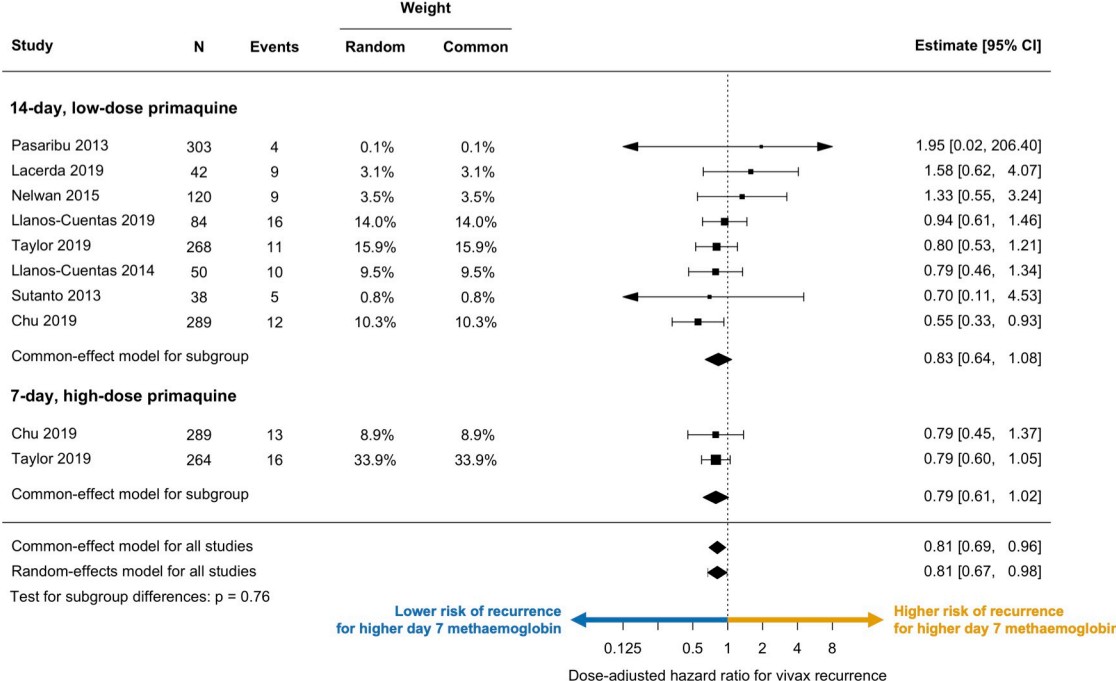

**Fig 5. Forest plot of the relationship between day 7 methaemoglobin levels and the risk of vivax recurrence.** Dashed line represents no association (i.e., hazard ratio = 1). Estimates to the left of the dashed line indicate a lower risk of recurrence with higher day 7 methaemoglobin levels (blue arrow). Conversely, estimates to the right indicate a higher risk of recurrence with higher day 7 methaemoglobin levels (orange arrow). Both the common-effect and random-effects models yielded comparable estimates. Horizontal axis represents dose-adjusted hazard ratios for vivax recurrence associated with a doubling in day 7 methaemoglobin, and is displayed on a logarithmic scale. CI confidence interval, N number of patients.

Our findings are in line with recent estimates derived from Chu and colleagues [10] (a secondary analysis of primaquine trial [9] data from the endemic northwest Thailand-Myanmar border) and Watson and colleagues [4] (an individual patient data meta-analysis of tafenoquine trials). These estimates suggested that an increase of 1 percentage-point of day 7 methaemoglobinaemia was associated with an estimated reduction of 10% for primaquine and 20% for tafenoquine in the risk of *P. vivax* recurrence (Fig J in S1 Text provides a graphical comparison). While our analysis of primaquine included data from these studies such that estimates are partially correlated, additional data from different studies contributed most (nearly 70%) of our pooled data set. Watson and colleagues [4] analysed vivax patients who received tafenoquine only (not included in this analysis). According to the White and colleagues review [8] of early experiments with primaquine or its analogues conducted more than 70 years ago [38–40], 8-aminoquinoline analogues which resulted in less than 6% of methaemoglobinaemia during treatment showed reduced efficacy. All these results point towards higher methaemoglobin levels following treatment as indicative of increased antihypnozoite activity at the population level.

In the current analysis of patients with normal G6PD activity, we observed no evidence of an association between the extent of early haemolysis on days 2 to 3 from baseline and day 7 methaemoglobinaemia. This suggests that during the early days of illness, early haemolysis was primarily attributable to the acute parasitaemia and rehydration, rather than iatrogenic haemolysis caused by the primaquine active metabolites. The lower day 7 methaemoglobin observed among younger patients may be partly explained by age-related enzyme immaturity and lower drug exposures [10]. The reason for lower methaemoglobin in older patients is less clear.

Methaemoglobinaemia is a simple and readily measured surrogate endpoint that can be quantified within a week after starting administration of an 8-aminoquinoline and has potential to improve the efficiency of exploratory trials (drug–drug interactions, regimen optimisation, drug screening, dose optimisation). Current antirelapse clinical trials generally require patient follow up for many months to observe recurrent infections in order to have sufficient power to determine comparative efficacy of different treatment arms. Quantifying methaemoglobinaemia also has potential to be a useful approach for monitoring of trials and clinical practices at an individual level, serving as a surrogate marker for patient adherence. Methaemoglobin could also serve as an adherence or exposure surrogate in clinical trials investigating radical curative efficacy (particularly for the investigation of relapses or unexpectedly poor radical curative efficacy). Although methaemoglobin is a surrogate endpoint for adherence to treatment, to our knowledge, these results are not driven by adherence (90% of patients had fully supervised treatment). Early-phase studies, preceding more definitive efficacy trials, may stand to benefit the most from this novel endpoint [8]. Minimum required sample sizes could also be substantially reduced by using day 7 methaemoglobin as an endpoint to improve cost-efficiency in conducting *P. vivax* trials. An ongoing study (NCT05788094) is currently using day 7 methaemoglobin measurement as a secondary outcome to address the question of whether there is important drug–drug interaction between tafenoquine and chloroquine, dihydroartemisinin (DHA)-piperaquine, or artemether-lumefantrine [41]. The recent INSPECTOR trial in Indonesia used DHA-piperaquine as the partner schizonticidal drug and lower than expected efficacy was observed [42] compared to trials with chloroquine [43–45]. In the INSPECTOR study, tafenoquine plus DHA-piperaquine resulted in a median methaemoglobin of 1.3% (range = 0.7% to 3.7%), which was similar to DHA-piperaquine alone (median = 1.0%, range = 0.5% to 1.8%). In contrast, primaquine plus DHA-piperaquine produced more than twice as much methaemoglobin (median = 2.9%, range = 0.9% to 7.9%) and superior efficacy than the tafenoquine plus DHA-piperaquine arm. While individual patient

data from this trial could not be incorporated into our pooled analysis due to substantial time for obtaining approval and standardisation, the results appear to corroborate our findings that higher increases in blood methaemoglobin corresponded to improved radical cure rates.

Our study has several limitations. Some patients missed measurements and some studies by design intentionally did not measure methaemoglobin on day 7. We addressed such missing data through linear interpolation, since methaemoglobin was recorded within 2 days before and/or after day 7. It is important to note that methaemoglobinaemia itself is an inherently noisy measure with substantial variation between patients, especially in patients receiving higher doses of 8-aminoquinoline treatment (Fig I in S1 Text shows higher methaemoglobin variability in higher dose groups). There is also a possibility of less accurate methaemoglobin measurements by using CO-oximetry for patients with darker skin [46]. Improved reliability may be achieved by measuring methaemoglobin on both hands, incorporating multiple or repeated measurements, waiting longer, and blocking fluorescent light (e.g., turning off lights or covering the hand when checking). More invasive methods, such as directly measuring the biomarker in capillary blood, could enhance measurement accuracy but are less acceptable and would be logistically challenging in low-resource settings. In an upcoming analysis, we plan to explore alternative summary metrics for methaemoglobin levels using pharmacometric modelling [25]. This approach aims to capture exposures to primaquine's active metabolites and improve the overall robustness of our findings.

Only 2 studies [47,48] were conducted in locations with negligible reinfection. Therefore, there is uncertainty regarding the aetiology of *P. vivax* recurrences for most of the patients, since there is currently no standardised method to differentiate between recrudescence, reinfection, and relapse [1]. It is important to note that the use of highly efficacious schizonticides across the study sites makes recrudescence unlikely. Another potential source of error includes participants who might not have had hypnozoites in the liver to start with, thus having zero risk of relapse, despite primaquine-inducing methaemoglobinaemia. However, this circumstance should bias the effect estimates towards the null (i.e., no association between day 7 methaemoglobinaemia and the risk of vivax recurrence), making our estimates conservative.

The generalisability of our findings is constrained to the large majority of vivax malaria patients with normal G6PD activity (≥30% G6PD activity or a negative qualitative test). For individuals with G6PD deficiency, blood methaemoglobin will not serve as a valid surrogate endpoint for antihypnozoite activity [8,10]. In G6PD deficiency, primaquine does not induce significant methaemoglobinaemia [15,49]. We currently lack sufficient data to explore how *CYP2D6* polymorphisms impact primaquine biotransformation to its active metabolites, as most patients were not genotyped [10]. The previous analysis by Chu and colleagues suggested lower day 7 methaemoglobin in null *CYP2D6* metabolisers (genotype-based activity score of 0) [8]. Intermediate metabolisers may also have lower active metabolites and higher relapse rates [6,7]. Further research specifically targeted at these vulnerable populations is needed.

Our pooled data set included patients from various countries with different antimalarial treatment policies for their first-line schizonticides, each having different elimination kinetics and thus different durations of suppressive posttreatment prophylaxis. The use of time-to-event models in such contexts may introduce bias in favour of drugs with longer half-lives, such as chloroquine. However, our sensitivity analysis, employing logistic regression that included partner drug as a covariate, yielded similar predictive-effect estimates. The goal of this analysis was not to establish day-7 methaemoglobin levels as an individual level predictor of relapse risk. Predicting relapse at the individual level is very difficult because there is large interindividual variation in liver hypnozoite burdens [1,50]. A patient may have no methaemoglobin response to primaquine but not subsequently relapse because they had no

hypnozoites at enrolment. Indeed, in no primaquine treated arms of the included studies, the risk of any recurrent illness by day 120 ranged from around 25% to 80%.

In conclusion, in primaquine-treated individuals, the day 7 methaemoglobin concentration is a pharmacodynamic proxy for exposure to the biologically active metabolites of primaquine, making it a valid population-level surrogate endpoint in patients with *P. vivax* malaria who have normal G6PD activity. The consistency with results observed in tafenoquine studies suggests a common drug-class phenomenon for 8-aminoquinolines. Direct comparisons of 8-aminoquinoline-induced methaemoglobinaemia between primaquine and tafenoquine in future studies could prove useful. These findings collectively enhance our understanding of the causal mechanisms by which 8-aminoquinoline drugs exert their effects, facilitate drug discovery and regimen optimisation, and influence clinical practices.

## Supporting information

**S1 Text. Supporting Information.** Systematic search terms for the databases. Signalling questions for risk of bias assessment using the QUIPS tool adapted to the current analysis. Table A. PRISMA-IPD checklist. Table B. Risk of bias assessment. Table C. Studies included in analysis. Table D. Studies that were eligible for analysis but not included in the pooled data. Table E. Comparison of characteristics of patients (as originally reported) who received primaquine between included and eligible but not included studies. Table F. Regression table output for the main Cox proportional hazards model. Fig A. Study sites that contributed to the pooled data in this individual patient data meta-analysis. Fig B. Distribution of weight-adjusted primaquine daily dose by primaquine regimen. Fig C. Dynamics of primaquine-induced increases in blood methaemoglobin over time. Fig D. Day 7 methaemoglobin concentrations by primaquine regimen and dose group. Fig E. Inverse J-shaped association between patient age and day 7 methaemoglobin by recurrence status, after controlling for daily mg/kg primaquine dose. Fig F. Distribution of day 7 methaemoglobin among the patients on (A) the original scale and (B) the logarithmic scale. Fig G. Mixed-effects estimates for the intercept and slope (mean difference). Fig H. Mixed-effects estimates for the slope (hazard ratio). Fig I. Example of sample size calculations for future studies. Fig J. Comparisons of estimates from the current analysis with those previous studies.
(DOCX)

## Acknowledgments

We thank all patient volunteers, healthcare workers, and research staff who contributed to the individual studies. We thank the WWARN team for technical and administrative support.

## Author Contributions

**Conceptualization:** Ihsan Fadilah, Robert J. Commons, Ric N. Price, Nicholas J. White, J. Kevin Baird, James A. Watson.

**Data curation:** Ihsan Fadilah, Robert J. Commons.

**Formal analysis:** Ihsan Fadilah, Robert J. Commons, James A. Watson.

**Funding acquisition:** Ihsan Fadilah, Nicholas J. White, J. Kevin Baird.

**Methodology:** Robert J. Commons, Nicholas J. White, J. Kevin Baird, James A. Watson.

**Project administration:** Ihsan Fadilah.

**Resources:** Robert J. Commons, Nguyen Hoang Chau, Cindy S. Chu, Nicholas P. J. Day, Gavin C. K. W. Koh, Justin A. Green, Marcus VG Lacerda, Alejandro Llanos-Cuentas, Erni J. Nelwan, Francois Nosten, Ayodhia Pitaloka Pasaribu, Inge Sutanto, Walter R. J. Taylor, Kamala Thriemer, Ric N. Price, Nicholas J. White, J. Kevin Baird, James A. Watson.

**Supervision:** Robert J. Commons, Ric N. Price, Nicholas J. White, J. Kevin Baird, James A. Watson.

**Visualization:** Ihsan Fadilah.

**Writing – original draft:** Ihsan Fadilah, Robert J. Commons, Ric N. Price, Nicholas J. White, J. Kevin Baird, James A. Watson.

**Writing – review & editing:** Ihsan Fadilah, Robert J. Commons, Nguyen Hoang Chau, Cindy S. Chu, Nicholas P. J. Day, Gavin C. K. W. Koh, Justin A. Green, Marcus VG Lacerda, Alejandro Llanos-Cuentas, Erni J. Nelwan, Francois Nosten, Ayodhia Pitaloka Pasaribu, Inge Sutanto, Walter R. J. Taylor, Kamala Thriemer, Ric N. Price, Nicholas J. White, J. Kevin Baird, James A. Watson.

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
