## [Editor Report · Decision Letter 0]

7 May 2024

Dear Dr Fadilah, 

Thank you for submitting your manuscript entitled "Methaemoglobin as a surrogate marker of primaquine antihypnozoite activity in Plasmodium vivax malaria: a systematic review and individual patient data meta-analysis" for consideration by PLOS Medicine.

Your manuscript has now been evaluated by the PLOS Medicine editorial staff and I am writing to let you know that we would like to send your submission out for external peer review.

Please re-submit your manuscript within two working days, i.e. by May 09 2024.

Feel free to email me at aschaefer@plos.org or us at plosmedicine@plos.org if you have any queries relating to your submission.

Kind regards,

Alexandra Schaefer, PhD

Associate Editor

PLOS Medicine

---

## [Decision Letter · Decision Letter 1]

17 Jun 2024

Dear Dr Fadilah,

Many thanks for submitting your manuscript "Methaemoglobin as a surrogate marker of primaquine antihypnozoite activity in Plasmodium vivax malaria: a systematic review and individual patient data meta-analysis" (PMEDICINE-D-24-01457R1) to PLOS Medicine. The paper has been reviewed by subject experts and a statistician; their comments are included below and can also be accessed here: [LINK]

As you will see, the reviewers were positive about the paper, but raised a number of questions about specific study details, clinical utility and the methodological approach. After discussing the paper with the editorial team and an academic editor with relevant expertise, I'm pleased to invite you to revise the paper in response to the reviewers' comments. We plan to send the revised paper to some or all of the original reviewers, and we cannot provide any guarantees at this stage regarding publication.

We ask that you submit your revision by Jul 08 2024. However, if this deadline is not feasible, please contact me by email, and we can discuss a suitable alternative.

Don't hesitate to contact me directly with any questions (atosun@plos.org). 

Best regards, 

Alex

Alexandra Tosun, PhD 

Associate Editor

PLOS Medicine

atosun@plos.org

Comments from the editorial team:

The editorial team feels that the point raised by reviewer #3 regarding the "substantial overlap in levels between individuals who suffer parasite recurrence and individuals who do not" is very important - please address this comment carefully and consider adding an analysis of the ability of day 7 methemoglobin to predict the odds of recurrence (or time to recurrence) at the individual level. We also ask that you further discuss clinical utility in the context of standard biomarker performance metrics. 

Comments from the reviewers: 

Reviewer #1: Timely conducted research project. The relapses of P vivax is one of the problem for elimination of malaria from countries. This has given an insight how to predict relapses. However, this needs to conduct in field level to see the practicability of measuring methemoglobin in low resource settings. Further, need to conduct future studies to determine dose and regime of primaquine/tafenoquine to obtain maximum level of methemoglobin.

Reviewer #2: See attachment

Michael Dewey

Reviewer #3: The authors have conducted systematic review and pooled analysis of individual participant data from 8 clinical trials across 9 geographical sites to study the association between day 7 meth-hemoglobin levels and the sub-sequent risk of recurrent parasitemia in individuals treated with primaquine (an 8-aminoquinoline) for radical cure of P. vivax hypnozoites.

The paper is very well written and methodologically sound. It provides important insights regarding day 7 meth-hemoglobin as a surrogate marker for treatment outcome as well as insights regarding the potential mechanisms by which 8-aminoquinolines act on dormant liver-stage parasites. The findings could have impact on the design future studies on drug efficacy.

The findings are of high interest for researchers studying the effect 8-anioquinolines and related compounds as well as for researchers conducting clinical trials on radical cure of dormant plasmodium liver-stage parasites and/or sexual stages.

My main concern is that although day 7 meth-hemoglobin appears promising as a surrogate marker of treatment outcome on average, there appears to be a substantial overlap in levels between individuals who suffer parasite recurrence and individuals who do not. To what extent 7 meth-hemoglobin levels can accurately predict treatment outcome on the individual level remains unclear from the results presented in the manuscript. I believe this is important to assess in order to evaluate the usefulness of day 7 meth-hemoglobin as a surrogate marker of treatment outcome. I would therefore like to ask the authors to complement the results with an analysis of the ability of day 7 meth-hemoglobin to predict the odds of recurrence (or time to recurrence) on the individual level.

Should the authors believe that this is not relevant, I believe this has to be explained and motivated clearly throughout the manuscript.

Apart from that I only have a few very minor suggestions:

Lines 436 - 449: In this paragraph the authors discuss variability in measurement of methhemoglobin particularly related to transcutaneaous measurment. It could be worth commenting whether or not if "invasive" measurement through analysis of capillary blood on e.g. on a blood-gas analyzer (also CO-oximetry based) could reduce measurement variability.

Figure legends - This is a general comment but figure legends do not specify what the trend lines in the different figures represent. Please clarify this to avoid any confusion.

Finally, I think it would be valuable if the authors included the regression table (at least for the main analysis based on the cox proportional hazards model) as supplementary information for the interested reader to scrutinize.

Reviewer #4: This article by Fadilah et al. describes a comprehensive review of literature on clinical trials that explored primaquine use to clear hypnozoite (dormant) forms of P. vivax. Beginning with 219 trials, the authors used well thought out criteria to narrow their analysis to 8 trials, containing 1,747 G6PD-normal patients spread across 8 countries. The goal was to compare day 7 methemoglobinemia levels with time to relapse. Their analysis showed that a doubling on day 7 of methemoglobin percentage was associated with an estimated 30% reduction in the risk of vivax recurrence following primaquine administration. The authors provide a compelling argument that measuring the methemoglobin levels on day 7 is far preferable, logistically and financially, to extending studies and assessing whether recrudescence occurred (with those trials typically extending up to 120 days). This makes logical sense. I should also note that many of the world leading malaria physicians and modelers working on malaria in SE Asia are on this manuscript. Overall, I found this to be an excellent study. It would have been clearly state textually how many trials would have enabled an analysis of tafenoquine. 

Early and cost-effective screening that would predict recrudescence in P. vivax-infected patients is an important topic. The authors have put together an excellent analysis. While this is a systematic review of trials data, and not a new trial itself, I find the topic and the rigor of the analysis a good fit for PLoS Medicine. The findings merit publication in this high-level journal. Although rare for me, I found nothing that required revision. 

Containing 

Reviewer #5: Reviewer's comments:

Summary: In this manuscript, the authors present a systematic data analysis of clinical trials involving the treatment with primaquine of P. vivax infected patients with the goal to validate methaemoglobin as a surrogate marker of anti-relapse efficacy of this 8-aminoquinoline drug. The authors applied a stringent selection of data from 8 studies out of 219 representing 1747 patients (out of 4122) in 24 study sites and 8 countries. The analysis, carefully conducted by the authors, shows that an increase of the primaquine dose correlates with an increase of methaemoglobin. A 2-fold increase of methaemoglobin percentage in blood of patients is associated with a 30% decrease in the risk to see P. vivax relapsing in such patients, thus validating methaemoglobin as a surrogate marker of the anti-relapse efficacy of primaquine.

The study is well conducted and clearly reported in this manuscript. The method is well described, and figures are self-explanatory.

I am very supportive of the publication of this manuscript in PLoS Medicine, provided that the authors adequately address the points raised below.

Major comments:

1) The authors should explain the statistical method they used to validate methaemoglobin as a surrogate of anti-relapse efficacy prognostic for primaquine after a stringent selection that consisted of keeping only 8 out of 219 clinical trials considered. The number of patients remaining in the analysis is certainly high enough to achieve the statistical power; however, it would be good to develop that aspect to further strengthen the manuscript.

2) Figure 1 is interesting but, in my opinion, should be the last one in the manuscript, supporting the discussion and integrating all the points discussed with CYP450-dependent metabolism of primaquine, possible absence of sporozoites yet detection of methaemoglobin, G6PD deficiency and absence of methaemoglobin (despite the fact that G6PD deficient patients are not considered in this study), etc… to provide readers with a more integrated view of the regulations of parasitemia and drug level of the metabolic intermediate in the liver associated to methaemoglobin.

Minor comments:

Lines 292: Figure 2 shows weight-adjusted… should read Figure S2.

Line 318: Figure 6 is mentioned but here again Figure S6 is the valid one.

Line 425 : …a day 7 methaemoglobin should be amended as follow: … a day 7 methaemoglobin measurement…

Legend of figure 4 last sentence: "Vertical axis is show on the logarithmic scale" should read: "Vertical axis is shown on the logarithmic scale.

---

* Please upload any figures associated with your paper as individual TIF or EPS files with 300dpi resolution at resubmission; please read our figure guidelines for more information on our requirements: http://journals.plos.org/plosmedicine/s/figures. While revising your submission, please upload your figure files to the PACE digital diagnostic tool, https://pacev2.apexcovantage.com/. PACE helps ensure that figures meet PLOS requirements. To use PACE, you must first register as a user. Then, login and navigate to the UPLOAD tab, where you will find detailed instructions on how to use the tool. If you encounter any issues or have any questions when using PACE, please email us at PLOSMedicine@plos.org.

* For authors with ties to industry, please indicate whether any of the interests has a financial stake in the results of the current study.

FIGURES AND TABLES

SUPPLEMENTARY MATERIAL

REFERENCES

STUDY TYPE-SPECIFIC REQUESTS

* Please report your SR/MA according to the PRISMA guidelines provided at the EQUATOR site. http://www.equator-network.org/reporting-guidelines/prisma/. Please provide the completed PRISMA checklist as Supporting Information. When completing the checklist, please use section and paragraph numbers, rather than page numbers. Please add the following statement, or similar, to the Methods: "This study is reported as per the Preferred Reporting Items for Systematic Reviews and Meta-Analyses (PRISMA) guideline (S1 Checklist)." 

* Abstract: Please report your abstract according to PRISMA for abstracts (https://doi.org/10.1371/journal.pmed.1001419) following the PLOS Medicine abstract structure (Background, Methods and Findings, Conclusions). Please ensure you provide dates of search, data sources, number of studies included, types of study designs included, eligibility criteria, and synthesis/appraisal methods.

* Please note that we expect searches to be updated to within 6 months of the time of submission.

---

## [Decision Letter · Decision Letter 2]

6 Aug 2024

Dear Dr. Fadilah,

Thank you very much for re-submitting your manuscript "Methaemoglobin as a surrogate marker of primaquine antihypnozoite activity in Plasmodium vivax malaria: a systematic review and individual patient data meta-analysis" (PMEDICINE-D-24-01457R2) for review by PLOS Medicine.

Thank you for your detailed response to the editors' and reviewers' comments. I have discussed the paper with my colleagues and the academic editor, and it has also been seen again by two of three original reviewers. The changes made to the paper were mostly satisfactory to the reviewers. As such, we intend to accept the paper for publication, pending your attention to the reviewers' and editors' comments below in a further revision. When submitting your revised paper, please once again include a detailed point-by-point response to the reviewers' and editors' comments.

[LINK]

In revising the manuscript for further consideration here, please ensure you address the specific points made by each reviewer and the editors. In your rebuttal letter you should indicate your response to the reviewers' and editors' comments and the changes you have made in the manuscript. Please submit a clean version of the paper as the main article file. A version with changes marked must also be uploaded as a marked up manuscript file. Please also check the guidelines for revised papers at http://journals.plos.org/plosmedicine/s/revising-your-manuscript for any that apply to your paper. 

We ask that you submit your revision within 1 week (Aug 13 2024). However, if this deadline is not feasible, please contact me by email, and we can discuss a suitable alternative.

Please do not hesitate to contact me directly with any questions (atosun@plos.org). If you reply directly to this message, please be sure to 'Reply All' so your message comes directly to my inbox.

We look forward to receiving the revised manuscript.

Sincerely,

Alexandra Tosun, PhD

Associate Editor 

PLOS Medicine

plosmedicine.org

Requests from Editors:

We noticed that you stated in the manuscript that you "included randomized therapeutic trials and prospective cohort studies published between January 1, 2000 and September 29, 2022". We are not sure if the above statement means that your last search was conducted in 2022, or if no studies were eligible after September 2022. Please note that we normally expect searches to be updated within 6 months of submission. However, we recognize that providing an update for individual patient data can be a significant undertaking. Therefore, we will not make this an editorial requirement in your case. To provide more transparency, we suggest that you perform an updated search and include in the limitations sections of the discussion how many potential studies could be included with an updated search.

We agree with Reviewer #3's comments regarding clarification that the aim of this study is to examine day 7 methemoglobin at the population level (rather than at the individual level). Please revise accordingly.

ABSTRACT

1) Please combine the Methods and Findings sections into one section.

2) l.71: Please define G6PD at first use.

3) l.71: “G6PD-normal patients” – please note that we prefer the use of patient-centered language (i.e. “patients with normal G6PD levels”). Please revise throughout the main manuscript accordingly.

4) l.76, please change to “P. vivax recurrence”.

5) l.76: Please define ‘CI’ at first use. 

6) l.76: We suggest changing the reporting to (i.e., removing the equal sign between "95% CI" and the values): (adjusted hazard ratio = 0.70; 95% CI [0.57, 0.86]; p = 0.0005). Please revise accordingly throughout the main manuscript.

7) In the last sentence of the Abstract Methods and Findings section, please describe the main limitation(s) of the study's methodology.

AUTHOR SUMMARY

Thank you for providing the Author Summary. Please note that the Author Summary should immediately follow the Abstract in your revised manuscript and not be submitted as a separate file.

1) The last bullet point under ‘Why was this study done?’ should become the first bullet point under ‘What did the researchers do and find?’.

2) “all available methaemoglobin and clinical data” – We suggest softening this statement.

3) Please change to: Patients with higher methaemoglobin levels a week after starting primaquine treatment appear to have a lower risk of recurrence.

4) Please change to: Day 7 methaemoglobin levels can potentially be used as a proxy for later vivax recurrence in exploratory trials, making it more efficient (e.g., fewer volunteers and resources required) to determine whether new drugs or regimens are effective.

5) In the final bullet point of 'What Do These Findings Mean?', please include the main limitations of the study in non-technical language.

INTRODUCTION

l.129: “all available data” - We suggest softening this.

METHODS AND RESULTS

1) Figure 1: Please define ‘P. vivax’ in the figure title.

2) l.273: Please change the heading to “Results”.

3) Figure 3: In the figure caption, please indicate the meaning of the bars and whiskers (e.g. median plus first quartile and third quartile).

4) l.325, please change to: The highest levels were observed among adolescent patients.

5) ll.325-326: “Younger and older patients had lower primaquine-induced methaemoglobinaemia (Fig S5).” – If possible, could you define "younger" and "older" by specifying an age (e.g., <5 years and >60 years)?

6) Figure 4: In the figure caption, please indicate the meaning of the bars, whiskers, and dots. For the top graph, please add a unit for the values on the x-axis. In addition, please describe what is shown in the two different graphs, especially the top graph.

7) Figure 5: Please define 'CI' and 'N' in the figure description. We suggest changing the figure title to be more specific. Please consider avoiding the use of red and green in order to make your figure more accessible to those with colour blindness.

8) Table 1: Please change the characteristic ‘Male’ to ‘Sex, Male’. We suggest changing the characteristic ‘Age (years)’ (the first one) to ‘Median Age (years)’ and ‘Primaquine daily dose (mg/kg)’ to ‘Median Primaquine daily dose (mg/kg)’. Please remove ‘ACT artemisinin-based combination therapy’ from the list of abbreviations since it is not used in the table.

DISCUSSION

1) l.391, please change to: “…we confirmed…”

2) l.436: Please temper claims of primacy of results by stating, "to our knowledge" or something similar.

3) l.441: Please define ‘DHA-piperaquine’ at first use.

REFERENCES

1) Where website addresses are cited, please specify the date of access using use the word “accessed” (e.g. [accessed: 10/04/2024]).

2) Please ensure that journal name abbreviations match those found in the National Center for Biotechnology Information (NCBI) databases (http://www.ncbi.nlm.nih.gov/nlmcatalog/journals), and are appropriately formatted and capitalised. For example, “The American Journal of Tropical Medicine and Hygiene” in reference [2] should be “Am J Trop Med Hyg”.

3) Please revise the supporting information references according to the comments above.

SUPPLEMENTARY MATERIAL

1) Thank you for providing the completed PRISMA-IPD checklist. Please replace the page numbers with paragraph numbers per section (e.g. "Methods, paragraph 1"), since the page numbers of the final published paper may be different from the page numbers in the current manuscript.

2) In the published article, supporting information files are accessed only through a hyperlink attached to the captions. For this reason, you must list captions at the end of your manuscript file. You may include a caption within the supporting information file itself, as long as that caption is also provided in the manuscript file. Do not submit a separate caption file.

When SI files are contained with a single file:

Please label the file as ‘S1 Supporting Information’.

Please apply alphabetical labelling to each table and figure contained within the S1 file. For example, ‘Fig A’ to ‘Fig Z’ and ‘Table A’ to ‘Table Z’.

Plain text does not need to be labelled and can just be given a title as necessary. For example, ‘Statistical Analysis Plan’.

Please cite tables/figures as ‘Fig A in S1 Supporting Information’ and/or ‘Table A in S1 Supporting Information’, for example.

Please cite plain text as, ‘Statistical Analysis Plan in S1 Supporting Information’, for example.

When SI files are uploaded as separate files:

Please label tables as ‘S1 Table’ (so on) and figures as ‘S1 Fig’ (and so on).

Any additional documents (protocols/analysis plans etc.) can be labelled as ‘S1 Protocol’, for example. Please cite items as exactly as labelled.

SOCIAL MEDIA

To help us extend the reach of your research, please provide any X (formerly known as Twitter) handle(s) that would be appropriate to tag, including your own, your co-authors’, your institution, funder, or lab. Please enter in the submission form any handles you wish to be included when we post about this paper.

Comments from Reviewers:

Reviewer #2: The authors have addressed my points

Michael Dewey

Reviewer #3: In the revised version of the manuscript the authors have largely addressed my concerns. However, I think it should be made evident in the abstract and ideally also in the manuscript title that the aim of this study is to explore day 7 methemoglobin as a population-level (rather than individual-level) marker of primaquine anti-hypnozoite activity. 

Reviewer #5: I consider that the points I raised after reading the first version of the submitted article were adequately addressed by the authors. In particular the clarification of the breakdown of the selected studies. consequently, I recommend the publication of this article in PLoS Med.

[LINK]

General Editorial Requests

---

## [Editor Report · Decision Letter 3]

27 Aug 2024

Dear Dr Fadilah, 

On behalf of my colleagues and the Academic Editor, James G. Beeson, I am pleased to inform you that we have agreed to publish your manuscript "Methaemoglobin as a surrogate marker of primaquine antihypnozoite activity in Plasmodium vivax malaria: a systematic review and individual patient data meta-analysis" (PMEDICINE-D-24-01457R3) in PLOS Medicine.

I appreciate your thorough responses to the reviewers' and editors' comments throughout the editorial process. We look forward to publishing your manuscript, and editorially there are only a few remaining minor stylistic/presentation points that should be addressed prior to publication. We will carefully check whether the changes have been made. If you have any questions or concerns regarding these final requests, please feel free to contact me at atosun@plos.org.

Please see below the minor points that we request you respond to:

1) Abstract: We suggest reverting the text (i.e., to September 29, 2022) and leaving it to the Methods section only to explain that an updated search to July 26, 2024 revealed two potential additional studies that were not included due to time constraints in obtaining approval and standardizing these data. We also suggest adding this point as a brief discussion in the limitations section of the Discussion. Alternatively, if you want to indicate in the Abstract that the search was conducted through July 26, 2024, you will need to update the main text, Abstract, and Figure 2 to indicate that you identified 221 studies and that two recent studies were excluded due to time constraints in obtaining approval and standardizing these data.

2) Figure 1: Please note that the figure title in lines 245/246 has not been updated to spell out P.vivax (it is correct in the figure title for Figure 1 in lines 775-777).

3) Figure 2: Pending on your answer to 1), please update the search dates and numbers and indicate in the figure that two studies were excluded due to time constraints in obtaining approval and standardizing the data.

4) Discussion: It appears that the definition of DHA has only been added to the text of the PDF file, but not to the track changes version. Please check.

5) Data sharing: Please update the Data Availability statement in the online submission form with the information provided on lines 586-596.

Before your manuscript can be formally accepted you will need to complete some formatting changes, which you will receive in a follow up email (including the editorial points above). Please be aware that it may take several days for you to receive this email; during this time no action is required by you. Once you have received these formatting requests, please note that your manuscript will not be scheduled for publication until you have made the required changes.

PRESS

Sincerely, 

Alexandra Tosun, PhD 

Associate Editor 

PLOS Medicine